# Health-Related Disparities among Migrant Children at School Entry in Germany. How does the Definition of Migration Status Matter?

**DOI:** 10.3390/ijerph17010212

**Published:** 2019-12-27

**Authors:** Amand Führer, Daniel Tiller, Patrick Brzoska, Marie Korn, Christine Gröger, Andreas Wienke

**Affiliations:** 1Martin-Luther-University Halle-Wittenberg, Institute of Medical Epidemiology, Biometrics and Informatics, 06112 Halle (Saale), Germany; Daniel.tiller@uk-halle.de (D.T.); marie.korn@posteo.net (M.K.); andreas.wienke@uk-halle.de (A.W.); 2Health Services Research Unit, Faculty of Health, School of Medicine, Witten/Herdecke University, 58448 Witten, Germany; Patrick.Brzoska@uni-wh.de; 3Public Health Department, City of Halle (Saale), 06112 Halle, Germany; Christine.Groeger@halle.de

**Keywords:** migration background, school entry examination, first-generation migration background, one-sided migration background, biopolitics

## Abstract

*Background*: Migration background is known to be an important risk factor for a number of medical outcomes. Still, relatively little is known about the epidemiologic relevance of different definitions of migration status. *Methods*: Data from 5250 school entry examinations spanning three consecutive years (2015–2017) were gathered from the Public Health Department in Halle, Germany. Data were stratified according to six different migration statuses and evaluated for differences in health service utilization and developmental outcomes. *Results*: Compared to non-migrant children, migrant children have a lower utilization of preventative services, and higher frequencies of developmental delays. Children with first-generation migration background consistently show results worse than all others, while children with one-sided second-generation migration background show results similar to those of their non-migrant peers. These findings are not substantially altered by adjustment for social status. *Conclusions*: Children with first-generation migration background should receive special attention in school entry examinations, since they constitute a group with consistently higher health risks compared to other groups of preschoolers.

## 1. Introduction

In many countries, large proportions of the population consist of individuals who were born abroad or whose parents were born abroad. While in the case of Luxembourg, this proportion is approximately 46%, in most EU-countries, it is between 10% and 20% of the population [1]. Similarly, the USA, Mexico or Australia report proportions of foreign-born people or their children of approximately 13% to 29% [2,3,4]. These individuals are commonly referred to as ‘migrants’, ‘immigrants’ or ‘people with a migration background’—although the definition of who actually belongs to this population group may differ across countries. 

Overall, migration background has been identified as an important determinant for many health-related outcomes all over the world [5,6]. Compared to the majority population, migrants have been shown to be disadvantaged in a number of health-related aspects, including health status [7,8,9], health behavior [10] and the utilization of health services [11,12,13]. 

In Germany, many migrants face obstacles in accessing health services [14], have a lower utilization of vaccinations and other preventative services [15], a higher prevalence of psychological and work-related complaints [16] and generally show worse health care outcomes than the majority population [17].

Since disparities do not only become evident among migrants of adult age but also among migrant children and adolescents, information on migration status is therefore considered to be relevant in pediatric checkups and school entry examinations (SEEs) [18]. 

However, there is a longstanding debate as to whether condensing different migration experiences into one common variable does justice to the heterogeneity of the population in question [7]. Critics argue that this generalizing view ignores the differences in patients’ health-related lifeworlds that come about by different migration experiences [19] and thereby imposes limits on sound analyses [20].

Therefore, in recent years, more complex definitions of migration status have been proposed that include not only the parents’ and the child’s nationality and country of origin but also factors like the language spoken at home or the parents’ mother tongue [21,22]. 

In Germany, the extent to which these suggestions are implemented in administrative routine procedures of SEEs varies between federal states [23] and the question as to whether the different definitions of migration status actually constitute groups of children with different risks for relevant health outcomes remains unanswered.

### 1.1. Question 

Using data from SEEs from the city of Halle (Saale), Germany, this present study investigates whether different definitions of migration status lead to different conclusions regarding disparities in (a) health service utilization (specifically, vaccinations and pediatric preventive check-ups), (b) the risk for relevant medical conditions diagnosed at school entry and (c) the recommendations given by health professionals at the end of a SEE. 

Thus, we seek to determine whether the *sociologically* sound distinction of different migration statuses is also *epidemiologically* relevant and whether the different definitions of migration status identify groups of children with different health needs.

In summary, the objective of this study is to assess how the definition of migration status matters within the context of disparities in health-related outcomes in SEE.

### 1.2. Theoretical Framework

The question as to whether different definitions of migration status are epidemiologically useful, needs to be further contextualized within the broader debate on one of epidemiology’s core activities: categorizing people with different risks for well-defined outcomes [24,25]. 

Many countries use an individual’s self-ascribed ethnicity to classify sociocultural background [26]. In Germany, most administrative procedures (including those within the health care system) do not offer the opportunity for self-ascription but rely on operationalized definitions based on administrative data—such as the child’s and the parents’ nationalities and countries of birth—as a starting point instead [27]. 

At first glance, using the resulting categories as variables for epidemiologic analyses seems to be unproblematic. Still, critics have been warning for decades that epidemiologists ought to consider the stigmatizing potential of such ascriptions [28,29,30,31]. Labels used to describe people are no inert variables, but can in turn play an important role in the “constitution of subjects” [32] and decisively shape people’s self-perception [33,34]. 

For this reason, such ascriptions should only be used if they are of undoubted importance and it can consequently be assumed that their usefulness outweighs their potential harm.

In addition, the statistical question, i.e., whether there are quantitative differences between the various definitions of migration status, this article therefore raises the question as to whether these differences are sufficiently pronounced to justify labelling children as “different”. 

## 2. Materials and Methods 

This study is an exploratory observational study that uses pooled secondary data from SEEs spanning three consecutive years for descriptive analysis. 

### 2.1. Dataset

In Germany, SEEs are obligatory examinations, where all children are examined by the respective communal departments of public health (*Gesundheitsamt*) in the year before starting school. These checkups comprise a parental questionnaire, a standardized examination including vision, hearing, and developmental tests, and a clinical examination by the public health physician that aim to detect developmental delays which could interfere with the child’s educational success [35,36,37,38], obtain epidemiological data [37,39,40,41] and provide individual medical help to children who are otherwise underserved [42,43,44].

For the purpose of this study, data on school entry examinations were obtained from the communal Public Health Department of Halle (Saale), Germany, for the years of 2015, 2016, and 2017. In these years, the Public Health Department examined 2056, 2013, and 2006 preschoolers respectively. During these respective years, parents did not agree to the use of data for secondary analysis in the cases of 319, 272 and 234 preschoolers; these children were excluded from analysis. Therefore, the dataset of all three years included information on a total of 5250 children. 

### 2.2. Variables

A child’s social status was estimated as ‘high’, ‘medium’ or ‘low’ according to the Brandenburg Social Index [45] using his or her parents’ level of education and employment status. 

Utilization of preventative health services was estimated from participation in preventative checkups and vaccination status. In Germany, there are a total of nine preventative pediatric checkups that are recommended from the age of one day (U1) to the age of five years (U9). Checkup participation was considered complete when the child took part in all nine checkups. Vaccination status for tetanus and measles was considered complete when a child received at least four doses of the tetanus vaccine and two doses of the measles vaccine before the SEE, as recommended by the Standing Committee on Vaccination [46]. 

The medical outcome variables and the recommendations derived from them were extracted from the documentation of the standardized testing and the clinical examination. As part of the Public Health Department’s routine procedure, the results of the standardized tests (based on standardized cut-off-values) are used to assess a child’s need for special aid called “early intervention”, for the postponement of school enrollment or for special educational care. Pathological results from the standardized tests for language (grammar and pronunciation), fine and gross motor skills, and cognition were used to derive the combined variable “severe developmental disorder” [47].

### 2.3. Definitions of Migration Status

Concerning a family’s migration experience, the parental questionnaire included information on the parents’ nationalities (at the moment of the SEE) and countries of birth, and on the child’s country of birth.

Using the information from the school entry examinations, we constructed six groups of migrant children based on six different definitions of migration status commonly used in literature: Migration status 1 represents the broadest definition of migration background, as it is used by the German Federal Statistical Office (*Statistisches Bundesamt*). Here, children are classified as migrants if they themselves *or* at least one of their parents had a nationality other than German at birth [27]. This definition of migration status is routinely used in SEEs in Saxony-Anhalt [47] and the German Federal Health Monitoring [48]. For a discussion of the historical development of this definition see [34]. 

Migration status 2 uses a narrower definition and is recommended in the so-called “basic set of indicators for mapping migration status” [21] and—specifically for the purpose of SEEs—by a working group of the conference of federal health ministers in Germany [22]. In this definition, status as a migrant is assigned if the child *and* at least one of the parents was born abroad, or if both parents were born abroad and/or have a nationality other than German at the time of the SEE. In contrast to migration status 1, this definition does not classify children as migrants if they were born in Germany and have only one parent with a non-German nationality. 

Migration statuses 3 and 4 go beyond the dichotomized ascription of migration status and further differentiate children according to the migrating generation: Children who experienced migration themselves are assigned to migration status 3 (‘first-generation migration background’) while migration status 4 comprises children whose family’s migration experience is limited to the parental generation (‘second-generation migration background’). This distinction is suggested by a number of government agencies [18,27] and is also used in the “German Health Interview and Examination Survey for Children and Adolescents (KiGGS)”, a survey on children’s health with more than 17,000 children [19,49]. 

Migration statuses 5 and 6 further differentiate children with a second-generation migration background according to their parents’ migration experience. Hereby, migration status 5 comprises all children whose parents were born abroad and/or hold a non-German nationality (‘two-sided second-generation migration background’), while migration status 6 includes children with only one parent of non-German nationality or country of birth (‘one-sided second-generation migration background’). This distinction is also recommended by the German Federal Statistical Office [27].

Non-migrants were defined as individuals who themselves and whose parents have German nationality and were born in Germany.

An overview of the migration statuses is given in Figure 1. 

### 2.4. Statistical Analysis

In a first step, we descriptively explored whether there are differences across migration statuses with respect to children’s demography, health care utilization, frequency of developmental disorders and recommendations received as a result of the SEE. For this purpose, data were stratified according to the aforementioned six migration statuses. Absolute and relative frequencies are reported. 

In a second step, prevalence ratios for each migration status were calculated to estimate the risk for the aforementioned endpoints. In order to adjust for possible confounding, the prevalence ratios were adjusted for socioeconomic status using logistic regression. 

Prevalence ratios—instead of odds ratios—were chosen as effect estimates, since some of the endpoints occur in the cohort with a prevalence of approximately 10 percent and thus the rare disease assumption might be violated [50]. 

Because of the descriptive character of the present investigation, no statistical tests were performed, as recommended by the American Statistical Association [51,52].

Analyses were performed using SAS^®^ (Version 9.4, Cary, NC, USA).

This study uses administrative data which fulfil all necessary requirements of the Federal Data Protection Act of the Federal Republic of Germany. As the data are fully anonymous and did not involve any experiments, according to national guidelines [53] no ethics clearance was necessary.

## 3. Results

### 3.1. Socio-Demographic Characteristics

A stratified analysis of children’s socio-demographic characteristics shows that irrespective of the definition of migration status that has been used, migrant children have a lower social status than non-migrant children. This trend is most pronounced for children with a first-generation migration background (migration status 3). Children with a one-sided second-generation migration background (migration status 6) are a notable exception here: They are more likely to have a higher social status than their non-migrant counterparts.

Similarly, they are also more likely to attend kindergarten. While children with second-generation migration background and two-sided second-generation migration background (migration statuses 4 and 5) show virtually identical frequencies of kindergarten attendance compared to non-migrant children, in the remaining groups migrant children are more often looked after at home and less often attend kindergarten. Again, this is most pronounced for children with first-generation migration background (migration status 3). 

Regarding family composition, marked differences between migrant and non-migrant families can be seen as well, again with one-sided second-generation migration background (migration status 6) being an exception: All other migration statuses are associated with a higher frequency of both parents living in the family and a smaller frequency of single mothers. Further, having more than two siblings is more frequent in migrant families.

The respective frequencies for socio-demographic details are shown in Table 1.

The countries of origin differ between the different definitions of migration status. For migration status 1 and 2, Germany is the most common country of birth for mothers, followed by Syria, Russia, Turkey and Vietnam. For fathers, Syria is most common, followed by Germany, Nigeria and Turkey. Children with first-hand migration experience (migration status 1) are mostly of Syrian origin. In families with only one foreign-born parent (migration status 6), it is in most cases the child’s father who migrated, with Nigeria being the most common country of father’s origin. More details concerning country of origin are presented in Table 2.

### 3.2. Health Service Utilization

When it comes to health service utilization, marked differences between migrant and non-migrant children become evident, as well as between the different definitions of migration status. While 12.6% of non-migrant children did not participate in the preventative checkup 8 (“U8”), in migrant children, this number was approximately 20%. Here, two exceptions are of note. Of those children with first-generation migration background (migration status 3), almost one-third missed the check-up, while children with one-sided second-generation migration background (migration status 6) show a U8-utilization similar to that of children without a migration background. For U9 and the overall completeness of all preventative checkups a similar picture emerges.

In terms of immunizations, the picture is less clear: Migrant children according to definitions 1 and 2 and children with second-generation migration background (migration status 4) have similar frequencies of complete tetanus vaccination status as their non-migrant peers, which is approximately 80%. In contrast, children with first-generation migration background (migration status 3) show a lower vaccination coverage (approximately 68%), while children with a two-sided second-generation migration background (migration status 5) are more often vaccinated (88%). For other vaccinations, the comparison of the migration statuses shows a similar picture. 

More details are to be found in Figure 2. 

### 3.3. Health Outcomes

Overall, the frequency of pathological findings in the SEE is higher in migrant children. Severe developmental disorders, mental retardation, language retardation and disorders of fine motor skills are all more prevalent among migrant children and are again most pronounced in children with a first-generation migration background (migration status 3). As for the utilization of checkups, children with one-sided second-generation migration background (migration status 6) are an exception to this trend and show frequencies similar to—or lower than—non-migrant children (see also Figure 3). 

This higher prevalence of developmental disorders among migrant children is contrasted with a lower prevalence of measures aimed at the treatment of diagnosed developmental disorders. While 3% of non-migrant children are already enrolled in early intervention programs at the time of the SEE, migrant children are enrolled in early intervention programs markedly less often (1.7%, 1.5%, 0%, 2.6%, and 1.2%, respectively); only children with one-sided second-generation migration background (migration status 6) take part in early intervention programs slightly more often (3.9%). Further, for migrant children, a postponement of school enrollment is proposed less often and they more rarely receive a recommendation for special educational support due to special needs.

The respective numbers are given in Figure 4.

### 3.4. Multivariate Analyses

Regression analyses show elevated risks for incomplete preventative checkups, incomplete vaccination status and severe developmental delays for all definitions of migration status except one-sided second-generation migration background (migration status 6). Children in this category have risks similar to the population of non-migrants, while—as in the descriptive analysis—first-generation migration background (migration status 3) is associated with the biggest health risks. When adjusting for social status the effects of migration statuses 1 to 5 are only slightly reduced. A notable exception here is first-generation migration background (migration status 3) where the risk for having missed preventive checkup U8 *increases* after adjustment for social status.

When it comes to the recommendations given at the end of the SEE, the picture is less clear. While children with all migration backgrounds are less likely to receive the recommendation for special educational care or early intervention, most of the effects are rather small and many confidence intervals include the null value. Interestingly, the risk for receiving no recommendations after the SEE increases for migrant children after adjusting for social status. 

Crude and adjusted prevalence ratios are displayed in Table 3. 

## 4. Discussion

In general, our findings are in line with other studies that found migration background generally to be a risk factor for lower health service utilization [19,54] or developmental delays [41,55]. 

Further, other studies that further differentiate migrant children find differences between groups of migrant children: A recent study from Hannover Region, Germany, distinguished preschoolers according to their parents’ country of origin and found that risk factors (in this case: levels of obesity) differ between non-migrant children and children whose parents have non-German nationalities, with children from Turkish families having the highest risk [56]. Studies from Australia reach similar conclusions. Here, children whose parents originate from the Pacific Islands and the Middle East have the highest risk [57,58]. Similar patterns depending on parents’ country of origin also appear in a systematic review on childhood obesity in the UK [59]. 

Few studies distinguish between different sub-categories of migration statuses. A study that analyzed the findings of SEE in Bavaria, Germany, used definitions of migration status similar to the ones employed here and also found children with migration background to generally fare worse in medically relevant outcomes compared to non-migrants [18]. As in our findings, children with one-sided migration background showed much better results than those with two-sided migration background and were more similar to non-migrant children in most aspects.

Similar results are found in the KiGGS-study, where children with first-generation migration background and two-sided second-generation migration background have a lower participation rate in pediatric checkups and children with one-sided migration background are very similar to non-migrant children [19].

Our research adds to this knowledge by showing that children with migration background have not only a higher risk for deficient utilization of preventative services and developmental delays, they are also less frequently prescribed early interventions and other measures which could potentially improve their development. 

This gap might be explained by the fact that none of the developmental tests used in the SEE in Halle (Saale) are administered in languages other than German and they are also not explicitly designed to be language-independent. As a result, we hypothesize that physicians might not trust the pathological results in children with low German-language proficiency and refrain from offering them the help a non-migrant child with similar test results would receive. This raises the question as to whether the SEE can provide reliable results for migrant children and highlights the need for language-independent tests. The urgency of this situation becomes more apparent when considering that migrant children constitute (depending on the definition) up to approximately one fifth of the age cohort.

Other relevant findings are the striking differences between children with first-generation migration background and children with one-sided second-generation migration background compared to non-migrant children, respectively. Even though all children with migration statuses 1 to 5 show increased health risks, children who migrated themselves show even worse results. Their lower utilization of preventative checkups and vaccinations might be explained by the recency of their migration to Germany (they simply were not in the country at the time the checkup or vaccination was due), or by parents’ health literacy which might not align well with the German health care system.

Still, the very high prevalence of language retardation emphasizes the need for better and earlier speech therapy interventions—or simply language instruction, depending on how the scores foreign speakers attain in language-dependent tests are interpreted. 

In contrast, children with one-sided second-generation migration background show results more similar to non-migrant children than to the other groups of migrant children. 

## 5. Limitations

Research on the influence of acculturation on health and health service utilization suggests that the duration of a family’s residence in Germany and the language spoken in the family might partly explain the differences observed in the present study [60,61]. Since this information was not available for our cohort, we cannot estimate its importance to our data. Similarly, it is well established that a family’s legal status may considerably affect children’s health, since precarious legal titles—like that of an asylum seeker or an undocumented migrant—substantially increase the risk for a number of adverse medical outcomes [8,62,63]. Since our data contained no information on legal status, its influence could not be accounted for in this paper. 

## 6. Conclusions

In summary, we conclude that migration statuses 1, 2, 3, 4 and 5 are all similarly associated with a higher risk for health-related findings in SEE compared to children without migration background. Children with first-generation migration background (migration status 3) show even higher risks and should therefore receive special attention in SEE, while children with one-sided second-generation migration background (migration status 6) perform similarly to children without migration background in school entry examinations. 

We therefore suggest that children with one-sided second-generation migration background should not be classified as migrants, since their socio-demographic characteristics and their risk for relevant medical findings in the SEE do not justify distinguishing them as a risk group separate from the general population of preschoolers. In contrast, children with first-generation migration background should receive special attention, since their risk for deficient utilization of preventative services and developmental delays is much higher compared to that of their non-migrant peers as well as to that of children with other migration statuses. 

## Figures and Tables

**Figure 1 ijerph-17-00212-f001:**
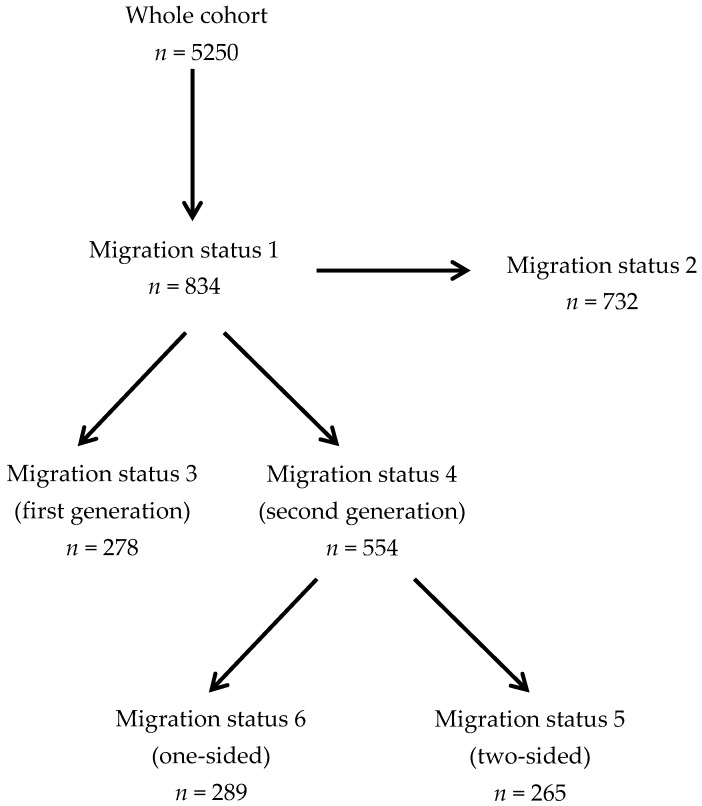
Overview over the different definitions of migration status.

**Figure 2 ijerph-17-00212-f002:**
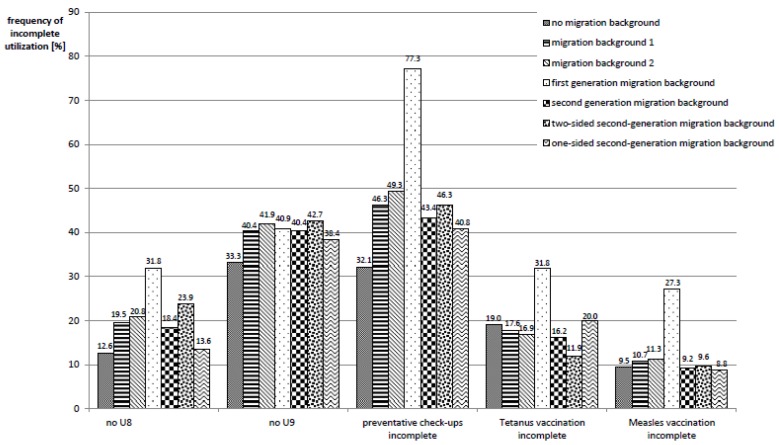
Frequencies of incomplete utilization of preventative health services, stratified according to migration status.

**Figure 3 ijerph-17-00212-f003:**
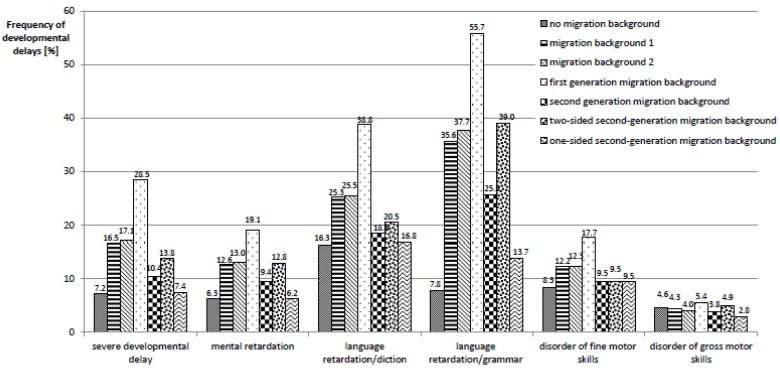
Frequencies of developmental delays, stratified according to migration status.

**Figure 4 ijerph-17-00212-f004:**
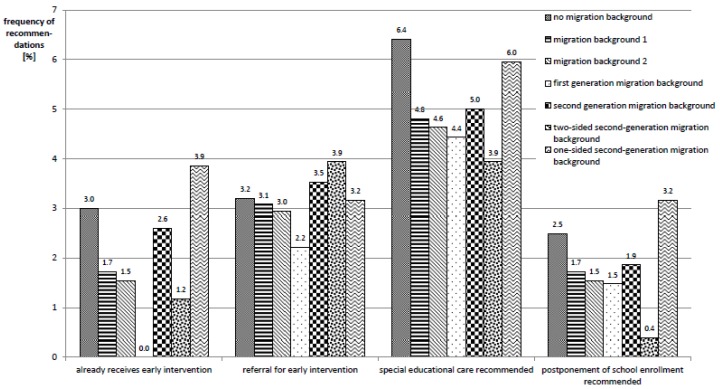
Frequencies of recommendations in relation to developmental delays, stratified according to migration status.

**Table 1 ijerph-17-00212-t001:** Socio-demographic characteristics of the study population.

	Migration Status 1		Migration Status 2		Migration Status 3		Migration Status 4		Migration Status 5		Migration Status 6	
	No	Yes	No	Yes	No	Yes	No	Yes	No	Yes	No	Yes
Sex												
Male	50.5%	49.8%	50.7%	48.2%	50.4%	50.7%	50.5%	49.3%	50.4%	50.2%	50.5%	48.5%
Female	49.5%	50.2%	49.3%	51.9%	49.6%	49.3%	49.5%	50.7%	49.6%	49.8%	49.5%	51.5%
Age (years)												
4	7.7%	6.6%	7.6%	6.9%	7.6%	6.7%	7.7%	6.5%	7.7%	4.8%	7.6%	7.8%
5	84.8%	85.8%	84.8%	86.2%	85.0%	84.7%	84.9%	86.3%	84.8%	89.9%	84.8%	83.6%
6	7.4%	7.6%	7.5%	6.9%	7.4%	8.6%	7.4%	7.2%	7.5%	5.3%	7.5%	8.6%
Socio-Economic Status *												
High	46.4%	38.5%	46.7%	35.4%	46.2%	24.9%	46.4%	44.6%	45.5%	38.7%	45.3%	49.3%
Medium	34.9%	28.3%	34.7%	29.3%	34.4%	25.4%	34.9%	29.7%	34.3%	28.0%	34.5%	31.0%
Low	18.7%	33.1%	18.7%	35.4%	19.5%	49.8%	18.7%	25.7%	20.2%	33.3%	20.3%	19.8%
Type of Child Care												
Kindergarten	96.8%	85.1%	96.6%	84.9%	96.8%	60.3%	96.8%	96.0%	95.2%	94.7%	95.0%	97.0%
At home	3.0%	14.7%	3.2%	15.0%	3.1%	39.2%	3.0%	4.0%	4.7%	5.3%	4.8%	3.0%
Nanny	0.2%	0.2%	0.2%	0.2%	0.1%	0.5%	0.2%	.	0.2%	.	0.2%	.
Parents’ Marital Status												
No information	0.2%	0.3%	0.2%	0.3%	0.2%		0.2%	0.4%	0.2%	1.0%	0.2%	.
Single mother	22.4%	18.4%	22.2%	19.0%	22.3%	10.5%	22.3%	21.9%	22.2%	12.1%	21.8%	29.5%
Mother and new partner	6.3%	3.2%	6.2%	3.0%	6.0%	1.9%	6.3%	3.8%	6.0%	1.5%	6.0%	5.6%
Both parents	68.6%	76.8%	68.9%	76.3%	67.0%	87.6%	68.6%	72.0%	69.1%	85.0%	69.5%	61.9%
other	2.6%	1.3%	2.6%	1.4%	2.6%		2.6%	1.9%	2.5%	0.5%	2.5%	3.0%
Number of Siblings												
<3	88.3%	80.2%	88.2%	79.5%	87.7%	75.1%	87.7%	82.3%	87.3%	83.1%	87.4%	81.7%
≥3	11.7%	19.9%	11.8%	20.5%	12.3%	24.9%	12.4%	17.7%	12.7%	16.9%	12.6%	18.3%

* Brandenburg Social Index according to [45].

**Table 2 ijerph-17-00212-t002:** Parents’ country of origin.

Mother’s Country of Origin	Migration Status 1	Migration Status 2	Migration Status 3	Migration Status 4	Migration Status 5	Migration Status 6
Germany	206 (24.8%)	159 (21.8%)	16 (5.8%)	190 (34.4%)	1 (0.4%)	189 (65.6%)
Syria	150 (18.0%)	144 (19.7%)	119 (42.8%)	31 (5.6%)	31 (11.7%)	0
Russia	48 (5.8%)	42 (5.8%)	6 (2.2%)	42 (7.6%)	27 (10.2%)	15 (5.2%)
Turkey	42 (5.1%)	39 (5.3%)	2 (0.7%)	40 (7.2%)	40 (15.1%)	0
Vietnam	41 (4.9%)	41 (5.6%)	2 (0.7%)	39 (7%)	33 (12.5%)	6 (2.1%)
Poland	24 (2.9%)	21 (2.9%)	11 (4%)	13 (2.4%)	3 (1.1%)	10 (3.5%)
Ukraine	22 (2.6%)	20 (2.7%)	6 (2.2%)	16 (2.9%)	13 (4.9%)	3 (1%)
Iraq	19 (2.3%)	16 (2.2%)	1 (0.4%)	18 (3.3%)	17 (6.4%)	1 (0.4%)
Kosovo	18 (2.2%)	18 (2.5%)	2 (0.7%)	16 (2.9%)	15 (5.7%)	1 (0.4%)
Kazakhstan	15 (1.8%)	11 (1.5%)	1 (0.4%)	14 (2.5%)	11 (4.2%)	3 (1%)
Father’s Country of Origin						
Syria	158 (19%)	149 (20.5%)	120 (43.3%)	38 (6.9%)	32 (12.1%)	6 (2.1%)
Germany	110 (13.3%)	85 (11.7%)	22 (7.9%)	88 (16%)	1 (0.4%)	87 (30.4%)
Nigeria	56 (6.8%)	51 (7%)	3 (1.1%)	53 (9.6%)	8 (3%)	45 (15.7%)
Turkey	54 (6.5%)	51 (7%)	4 (1.4%)	50 (9%)	41 (15.5%)	9 (3.2%)
Russia	35 (4.2%)	26 (3.6%)	3 (1.1%)	32 (5.8%)	23 (8.7%)	9 (3.2%)
Vietnam	34 (4.1%)	34 (4.7%)	1 (0.4%)	33 (6%)	33 (12.5%)	0
Iraq	32 (3.9%)	29 (4%)	1 (0.4%)	31 (5.6%)	17 (6.4%)	14 (4.9%)
Kosovo	19 (2.3%)	19 (2.6%)	1 (0.4%)	18 (3.3%)	16 (6%)	2 (0.7%)
Ukraine	17 (2.1%)	16 (2.2%)	1 (0.4%)	16 (2.9%)	12 (4.5%)	4 (1.4%)
India	16 (1.9%)	15 (2.1%)	10 (3.6%)	6 (1.1%)	2 (0.8%)	4 (1.4%)

**Table 3 ijerph-17-00212-t003:** Risks for severe developmental disorders according to migration status.

	No Preventative Checkup U8		Incomplete Tetanus Vaccination		Severe Developmental Delay		Special Educational Care Recommended		Referral for Early Intervention	
	Crude PR * (95%-CI)	Adjusted PR *^,†^ (95%-CI)	Crude PR * (95%-CI)	Adjusted PR *^,†^ (95%-CI)	Crude PR * (95%-CI)	Adjusted PR *^,†^ (95%-CI)	Crude PR * (95%-CI)	Adjusted PR *^,†^ (95%-CI)	Crude PR * (95%-CI)	Adjusted PR *^,†^ (95%-CI)
Migration status 1	1.57 (1.30–1.88)	1.57 (1.29–1.90)	2.14 (1.93–2.37)	2.02 (1.81–2.26)	2.22 (1.86–2.65)	1.57 (1.29–1.90)	0.76 (0.55–1.03)	0.59 (0.42–0.82)	0.96 (0.64–1.45)	0.74 (0.47–1.17)
Migration status 2	1.66 (1.38–2.01)	1.61 (1.31–1.98)	2.25 (2.03–2.49)	2.13 (1.9–2.38)	2.32 (1.93–2.77)	1.58 (1.30–1.93)	0.72 (0.51–1.01)	0.56 (0.39–0.79)	0.93 (0.6–1.44)	0.69 (0.43–1.13)
Migration status 3	2.35 (1.51–3.65)	3.05 (1.97–4.73)	3.83 (3.49–4.21)	3.56 (3.2–3.96)	3.69 (3.01–4.54)	2.16 (1.73–2.69)	0.75 (0.45– 1.27)	0.44 (0.25–0.79)	0.78 (0.37–1.64)	0.42 (0.18–1.02)
Migration status 4	1.49 (1.23–1.82)	1.47 (1.2–1.81)	1.28 (1.09–1.5)	1.31 (1.1–1.55)	1.47 (1.15–1.89)	1.14 (0.87–1.51)	0.76 (0.52–1.1)	0.68 (0.46–1.00)	1.06 (0.66–1.7)	0.95 (0.58–1.58)
Migration status 5	1.85 (1.45–2.35)	1.77 (1.36–2.3)	1.25 (0.99–1.56)	1.28 (0.99–1.64)	2.05 (1.54–2.74)	1.41 (1.02–1.95)	0.62 (0.35–1.12)	0.52 (0.29–0.96)	1.17 (0.62–2.19)	0.80 (0.38–1.68)
Migration status 6	1.19 (0.89–1.59)	1.22 (0.91–1.65)	1.31 (1.06–1.61)	1.34 (1.08–1.66)	0.94 (0.62–1.42)	0.85 (0.54–1.34)	0.88 (0.55–1.42)	0.85 (0.52–1.38)	0.96 (0.5–1.87)	1.1 (0.57–2.1)

* PR = prevalence ratio; reference group: children without migration background. ^†^ adjusted for socioeconomic status.

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
