# Peer review of "Health-Related Disparities among Migrant Children at School Entry in Germany. How does the Definition of Migration Status Matter?"

_ijerph, 2019, doi:10.3390/ijerph17010212_

Round 1
Reviewer 1 Report
It is an interesting piece of work about the association between immigration status and health disparities/risks in immigrants of a German city. Introduction: The introduction is solid and well posed. It is missing the general epidemiological data from Europe and Germany that support the problem. Provide more information about SEE. References 3, 5, 6 are very old and should be updated. The authors should provide more information about the definitions of immigration status by expanding the bibliography. The research question should focus on the search for association between immigration status and health disparities. The importance of the relationship between social and epidemiological variables is well described, looking at whether or not the relationship is relevant to justify labelling it. What do the authors mean by “making of people” (p. 2, l. 79)? It is missing a sentence that states: the objective of this study is… Method: Define the type of study. If it is an observational, descriptive and transversal design, specify it clearly in the beginning of the section. The variables are well defined, however, the definition of immigration status (p. 3, l. 146-142) lacks clarity. It appears to be a theory of the authors themselves supported by the diverse bibliography. Results: The results are interesting, well described, respond to the objectives and reference the tables. The health status of the immigrants in status 3 differs significantly in the measured variables with respect to non-immigrants; immigrant results in status 6 are similar to non-immigrants. Regression analysis shows greater risk in all immigrant statuses except in status 6 (similar to non-immigrants). Discussion: Interesting but very limited. The data is discussed with a study in Germany, which is insufficient. Contrast the results with similar studies in other European countries that also receive immigrants. Similarly, the cultural nature of the differences found among children is barely discussed. While this is included as a limitation, it is a major failure of the article. Conclusion and implications for practice are correct.
Author Response
Dear Sir or Madam
We would like to thank the reviewers for their helpful comments and the editors for the opportunity to revise our article.
We had the article language edited and adjusted the format of the references to the journal’s guidelines.
Below you find our comments concerning the details of the revision:
The introduction “is missing the general epidemiological data from Europe and Germany that support the problem.”We added more data on the demographic relevance of migrant populations throughout the world and specified some of the health disparities described for migrant populations worldwide and in Germany.
“Provide more information about SEE.”We added a paragraph to the methods section that outlines some more details concerning school entry examinations and references the respective literature.
“References 3, 5, 6 are very old and should be updated.”We substituted and/or amended the older literature by newer publications.
“What do the authors mean by “making of people” (p. 2, l. 79)?”“Making Up People” is the title of an influential text by the epistemologist Ian Hacking (1983), where he laid the foundation for a constructionist critique of epidemiology. To avoid interrupting the reading flow of readers not familiar with this debate, we removed the mention of Hacking’s catch-phrase from the text.
“It is missing a sentence that states: the objective of this study is…”We added this sentence under the heading “1.1 Questions”.
“Define the type of study. If it is an observational, descriptive and transversal design, specify it clearly in the beginning of the section.”We added at the beginning of the methods section that this is an observational study based on secondary data using exploratory analyses.
“The variables are well defined, however, the definition of immigration status (p. 3, l. 146-142) lacks clarity. It appears to be a theory of the authors themselves supported by the diverse bibliography.”Our construction of the groups of migrant children used all those definitions of migration status that we found in the literature and could ascribe to the children in our database with the variables available.
We slightly changed the wording of the introductory sentence under the heading “2.3” to better account for this approach and added some more references to the literature on the definitions of migration status.
Discussion: “Interesting but very limited. The data is discussed with a study in Germany, which is insufficient. Contrast the results with similar studies in other European countries that also receive immigrants.”We thank the reviewer for this hint. We added more articles on the topic to the discussion and contrast our findings with more studies using a similar approach.
“Similarly, the cultural nature of the differences found among children is barely discussed. While this is included as a limitation, it is a major failure of the article.”We understand this comment to hypothesize that the differences we observe between migrant children and their non-migrant peers are caused (or aggravated) by their cultural background.
We agree that this might be an important factor in addition to health care system-related mechanisms and added a sentence and references concerning this aspect in the discussions section.
Reviewer 2 Report
A Brief Summary
The study investigates whether different definitions of migration status lead to different conclusions regarding disparities in a) health service utilization (specifically, vaccinations and paediatric preventive check-ups), b) the risk for relevant medical conditions diagnosed at school entry and c) the recommendations given by health professionals at the end of a SEE. The authors endeavoured to determine if the sociologically sound distinction of different migration statuses is also epidemiologically relevant and if different definitions of migration status identify groups of children with different health needs.
The study findings imply that migration statuses 1, 2, 3, 4 and 5 are all similarly associated with a higher risk for health-related findings in SEE compared to children without migration background. Children with first generation background (migration status 3) show even higher risks and should therefore receive special attention in SEE, while children with one-sided second-generation migration background (migration status 6) are similar to children without migration background in school entry examinations
Broad Comments
The current study’s areas of strength include originality/novelty, significance, quality of presentation, and interest to the readers. The question is original and well defined as it is trying to address distinction of different migration statuses cross-disciplinary (i.e., epidemiologically and sociologically).
Also, the study’s significance is high as the conclusions are justified and supported by the results.
Quality of presentation is another strength of the study. The data and analyses are presented appropriately.
The areas of the study’s which could be improved relate to its scientific soundness, English level and overall merit (Please refer to specific comments for details).
Specific Comments
The aspect of the study which could be improved relates to its scientific soundness. Namely, it is unclear from the description of the statistical analyses, what software was used and whether the differences across migrations status were statistically significant. The authors noted that they descriptively explored if there are differences across migration status with respect to children’s demography, health care utilization, frequency of developmental disorders and recommendations. Although this was in line with the descriptive character of the study, perhaps comparing the mean values on respective variables could have provided a statistically stronger results and thus produce more nuanced results to shed light on potential policy implications.
Furthermore, it appears that the population could have been described in a more detailed manner, which could potentially increase the study’s external validity. Further information on sociodemographic characteristics of participants would inform possibilities of generalisation to other populations (e.g., in other federal states in Germany). Reporting on sociodemographic characteristics such as parents’ education level, country of origin, level of the knowledge of German language and length of stay in Germany, employment status, and access to health services seem to be related to the outcomes investigated in the study and would allow the authors to investigate how and to which extent these important variables influence health-related outcomes in migrants and non-migrants.
The overall merit of the study appears to be limited by the descriptive nature of the study design. It must be noted that the study provides an advance towards the current knowledge in that the authors are reassessing the definition of the migration status and its definition. Also, the finding that recommendations to migrant families have been offered less often than was the case with non-migrant families, highlights the importance of the study results and their potential policy implications. Nevertheless, one of the important limitations of the study is related to the fact that causal inferences between the investigated variables cannot be drawn based on the current study design.
Finally, the English language (spelling mistakes and grammar) seem to be one of the areas to be improved (e.g., line 40-44: sentence structure, line 48: English: grammar, line 53: sentence structure, line 62: paediatric, line 78: grammar, line 84-86: sentence structure, line 147: articles).
Author Response
Dear Sir or Madam
We would like to thank the reviewers for their helpful comments and the editors for the opportunity to revise our article.
We had the article language edited and adjusted the format of the references to the journal’s guidelines.
Below you find our comments concerning the details of the revision:
1. “The aspect of the study which could be improved relates to its scientific soundness. Namely, it is unclear from the description of the statistical analyses, what software was used and whether the differences across migrations status were statistically significant.”
“Nevertheless, one of the important limitations of the study is related to the fact that causal inferences between the investigated variables cannot be drawn based on the current study design.”
We certainly agree with the reviewer that the study design does not lend itself to causal inferences. Still, we do not think that performing significance tests would change this limitation.
In line with the recently published recommendations of the American Statistical Association (Wasserstein et al. 2019, The American Statistician 73, 1-19; Amrhein et al. (2019), Nature 567, 305-307) – which discourage the use of significance tests in exploratory approaches – we therefore would prefer not to follow the review’s advice in this matter. We added the references to these recommendations to the manuscript.
We added the information that analyses were done using SAS.
2. “Reporting on sociodemographic characteristics such as parents’ education level, country of origin, level of the knowledge of German language and length of stay in Germany, employment status, and access to health services seem to be related to the outcomes investigated in the study and would allow the authors to investigate how and to which extent these important variables influence health-related outcomes in migrants and non-migrants.”
We agree that these variables would be helpful for contextualizing our findings. Unfortunately, most of them are not routinely collected in SEEs and are therefore not available to us. We added information on the parents’ country of origin to the text and in a new table, Table 2.